# Recent Trends and Potential of Radiotherapy in the Treatment of Anaplastic Thyroid Cancer

**DOI:** 10.3390/biomedicines12061286

**Published:** 2024-06-10

**Authors:** Kazumasa Sekihara, Hidetomo Himuro, Soji Toda, Nao Saito, Ryoichi Hirayama, Nobuyasu Suganuma, Tetsuro Sasada, Daisuke Hoshino

**Affiliations:** 1Cancer Biology Division, Kanagawa Cancer Center Research Institute, Yokohama 2418515, Japan; sekihara.6e20l@kanagawa-pho.jp (K.S.); toda.soj.im@yokohama-cu.ac.jp (S.T.); saito.6d10l@kanagawa-pho.jp (N.S.); 2Biospecimen Center, Kanagawa Cancer Center, Yokohama 2418515, Japan; 3Division of Cancer Immunotherapy, Kanagawa Cancer Center Research Institute, Yokohama 2418515, Japan; himuro.4l00j@kanagawa-pho.jp (H.H.); sasada.0980e@kanagawa-pho.jp (T.S.); 4Department of Radiation Oncology, Kanagawa Cancer Center, Yokohama 2418515, Japan; 5Department of Breast and Thyroid Surgery, Yokohama City University Medical Center, Yokohama 2320024, Japan; 6Department of Charged Particle Therapy Research, QST Hospital, National Institutes for Quantum Science and Technology, Chiba 2638555, Japan; hirayama.ryoichi@qst.go.jp; 7Department of Surgery, Yokohama City University, Yokohama 2360004, Japan; suganuma@yokohama-cu.ac.jp

**Keywords:** anaplastic thyroid cancer (ATC), BRAF, MAPK signaling, PAM signaling, multimodality therapy, radiotherapy, molecular targeted therapy, immune checkpoint inhibitor

## Abstract

Anaplastic thyroid cancer (ATC) is a rare but highly aggressive malignancy characterized by advanced disease at diagnosis and a poor prognosis. Despite multimodal therapeutic approaches that include surgery, radiotherapy, and chemotherapy, an optimal treatment strategy remains elusive. Current developments in targeted therapies and immunotherapy offer promising avenues for improved outcomes, particularly for *BRAF*-mutant patients. However, challenges remain regarding overcoming drug resistance and developing effective treatments for *BRAF*-wild-type tumors. This comprehensive review examines the clinical and biological features of ATC, outlines the current standards of care, and discusses recent developments with a focus on the evolving role of radiotherapy. Moreover, it emphasizes the necessity of a multidisciplinary approach and highlights the urgent need for further research to better understand ATC pathogenesis and identify new therapeutic targets. Collaborative efforts, including large-scale clinical trials, are essential for translating these findings into improved patient outcomes.

## 1. Introduction

Although anaplastic thyroid cancer (ATC) occurs rarely [1,2,3,4], it exhibits an extremely poor prognosis [5,6,7] and is characterized by the presence of unresectable local extension or distant metastasis at initial diagnosis [8,9]. A radical cure for this malignancy is extremely difficult. To maintain and prolong quality of life (QOL), a multidisciplinary approach combining surgery, radiotherapy, and chemotherapy is currently used, but this approach yields poor results. Recently, clinical trials with kinase inhibitors have been conducted in cases in which specific genetic mutations, such as *BRAF* V600E, have been identified, with the hope of improving the prognosis of ATC. One example of a successful study was the combination of the BRAF inhibitor dabrafenib and the MEK inhibitor trametinib in ATC patients harboring the *BRAF* V600E mutation [10]. Certain reports have suggested the possible beneficial role of immune checkpoint inhibitors [7,8,9]. However, molecular targeted agents can lead to the emergence of drug resistance during administration, and this is a significant factor that complicates cancer treatment. Additionally, patients who do not possess the target gene mutation are not eligible for these agents. Immunotherapy can cause serious side effects, such as interstitial pneumonia, due to an excessive immune response when T cells in the body are activated.

Here, we summarize the clinical and biological features of ATC and the current standard of care and its problems, and we discuss future prospects for the management of ATC. Several useful review articles exist regarding ATC management [11,12,13,14,15], with emphasis on molecular targeted agents. We would like to distinguish our review from these other manuscripts by highlighting the potential for ATC management in the context of radiotherapy based on the remarkable technological development that has occurred in recent years.

## 2. The Characteristics of ATC

### 2.1. Epidemiology and Clinical Presentation

Most thyroid cancers are differentiated thyroid cancers (DTC) originating from follicular cells, such as papillary thyroid cancer (PTC) and follicular thyroid cancer (FTC), generally exhibit a good prognosis [16]. In contrast, ATCs are extremely aggressive and exhibit a poor prognosis. They are considered to be one of the most malignant of all cancer types [17,18] and account for 10–35% of thyroid cancer deaths, despite their low incidence (approximately 1–2% of all thyroid cancers) [2,4,19]. The majority of patients with this disease are older than 60 years of age, and the sex ratio is the lowest compared to other types of thyroid cancer that are more common in women. Patients present with rapidly growing cervical mass and neck pain that develops within weeks, skin erythema, hoarseness, dysphagia, dyspnea, rapid swelling of the cervical lymph nodes, fever, fatigue, and weight loss. Invasion of the gastrointestinal tract is frequently observed. Patients may occasionally develop urgent airway narrowing, such as invasion or compression of the trachea, requires airway clearance to prevent death due to asphyxia. According to the eighth edition of the American Joint Committee on Cancer (AJCC)/Tumor-Node-Metastasis (TMN) cancer staging system [20], the disease is always in an advanced stage (stage IV) and is divided into IVA (localized stage) when the cancer is only within the thyroid gland, IVB (locally advanced stage) when there is gross extra-thyroidal extension or cervical lymph node metastasis, and IVC (metastatic stage) when there is distant metastasis [7,21] (Table 1).

Onoda et al. [22] evaluated overall survival (OS) using this system in the national database of the Thyroid Cancer Research Consortium for Japan (ATCCJ) [23], which consisted of 757 patients with ATC. The percentages and median OS for each stage included stage IVA (5.9%, 15.8 months), stage IVB (55.7%, 6.1 months), and stage IVC (38.3%, 2.8 months). Consequently, approximately 90% of patients with ATC exhibited disease progression to surrounding organs or distant metastases at initial diagnosis, and treatment may not have been initiated due to extremely rapid disease progression. Moreover, it is common for patients with ATC to die early after diagnosis if no effective treatment is available. Older patients may not tolerate aggressive radiotherapy or chemotherapy due to impaired immune and/or organ function or complications. Thus, age is a significant prognostic factor for ATC [24]. Additionally, leukocytosis (white blood cell count >10,000/mL), extrathyroidal extension, and distant metastases have been reported as prognostic factors [23,25].

### 2.2. Genomic Changes

Recently, several analytical studies using next-generation sequencing have been conducted to investigate the genetic alterations in ATC [6,26,27,28,29,30,31,32,33]. According to their findings, the most commonly identified mutations in ATC were *TP53* and *telomerase reverse transcriptase (TERT)* promoter mutations. Although these mutations are observed in DTC and poorly differentiated DTC (PDTC), they are more frequent in ATC than they are in others [6,30]. *TP53* is a well-known tumor suppressor gene involved in a variety of cellular functions, including cell survival, DNA repair, apoptosis, cell cycle checkpoints, and senescence. *TP53* mutations promote cell proliferation and tumor progression through the loss of these functions [34]. This was strongly correlated with the malignant potential of ATC. Landa et al. [30] reported that *TP53* mutations were observed in 8% of PDTC; however, it increased to 73% in ATC. The analyses by Pozdeyev et al. [28] and Xu et al. [6] revealed high frequencies of *TP53* mutations (65% and 63%, respectively). *TERT* promoter mutations (C228T and C250T) are common in ATC. *TERT* is the catalytic subunit of telomerase and is essential for telomerase activity [35]. *TERT* is not expressed in most human somatic cells; however, its expression and transcription are upregulated in many cancers through various mechanisms, including mutations in the core promoter region of the *TERT* gene. Moreover, it elongates telomeres, confers unlimited proliferative capacity to tumor cells, and contributes to tumor progression and aggressiveness [36,37]. Recently, it has been reported that *TERT* interacts with various signaling molecules such as NF-κ B, c-MYC, β-catenin, and TCF-4 to increase cancer malignancy and promote cancer progression [38,39,40,41]. Landa et al. reported that *TERT* promoter mutations were observed at a high frequency of 73% in ATCs, compared to 9% in PTCs and 40% in PDTCs [30]. Xu et al. observed this mutation at a high frequency (75%) in ATCs [6]. 

Two major cascades are responsible for tumorigenesis in ATCs, including the mitogen-activated protein kinase (MAPK) signaling pathway (RAS-RAF-MEK-ERK) and the PI3K-Akt-mTOR (PAM) signaling pathway. As described in Figure 1, both pathways are involved in cell proliferation and survival and essentially start with receptor tyrosine kinase (RTK), which is activated by the binding of ligands such as growth factors such as fibroblast growth factor (FGF) or epidermal growth factor (EGF) to specific receptors. When one of these molecules is continuously activated by a genetic mutation, it consistently activates downstream molecules without signals from the upstream molecules.

The MAPK signaling pathway is involved in various cellular processes such as growth, proliferation, survival (avoidance of apoptosis [42] and induction of autophagy [43]), migration [44], and angiogenesis. The intracellular signaling of the MAPK pathway has been described in detail by Schubert et al. [45] and Cook et al. [46]. Common alterations in the MAPK pathway in ATC include mutations in *BRAF* and *RAS*. *BRAF* mutations are the primary therapeutic targets for this disease. The majority of *BRAF* mutations in ATC are *BRAF V600E* point mutations [29] that play an essential role in the development and progression of tumors [47,48]. This mutation activates BRAF kinase, which phosphorylates multiple targets, including MEK and ERK [49]. The *BRAF V600E* mutation is observed in approximately 20–40% of ATCs [50], and Jannin et al. [12]. reported that the frequency of *BRAF* mutations in ATCs appears to vary in each region. According to recent reports, *BRAF* mutations have been observed in 40–45% of ATC cases in the United States [6,30], 14–37% of cases in Europe [31,33], and 41–48% of cases in East Asia (Japan and Korea) [26,51]. Although the details are unclear, they suggest an association between ethnicity and iodine intake. *BRAF* mutations, the most common mutations originally observed in PTCs, strongly activate the MAPK pathway, ultimately leading to thyroid cell dedifferentiation [52]. Oishi et al. summarized the mutation profiles of PTCs and ATCs and observed that *BRAF* status and *BRAF* genotype matched between PTCs and ATCs in 18 of 21 cases [53]. Yoo et al. [26] and Xu et al. [6] reported similar results. Genomic associations have been demonstrated between PTCs and ATCs, thus suggesting that ATCs are derived from PTCs. *RAS* genes such as *HRAS*, *KRAS*, and *NRAS* have been described. *RAS* mutations are observed at a frequency of 10–50% in ATC, according to next generation sequencer (NGS) analysis data [27,28,54]. Among them, *NRAS* mutations are significantly more common in thyroid cancer [55]. Oishi et al. [53] reported *HRAS* and *KRAS* mutations at 5%, while *NRAS* mutations were detected at 18%. Lai et al. [56] confirmed a high frequency of *NRAS* (30%) compared to that of *HRAS* (0%) and *KRAS* (11%). *RAS* mutations promote the activation of the MAPK and PAM signaling pathways. *KRAS* mutants are major activators of the MAPK pathway, whereas *NRAS* mutants are activators of the PAM pathway [57]. These mutations were frequently detected in FTC [26], thus suggesting that a few ATC may have originated from FTC. Furthermore, certain studies have demonstrated that *eukaryotic translation initiation factor 1A X-linked (EIF1AX)* mutations often occur together with *RAS* mutations in ATCs and that a positive correlation between RAS and EIF1AX proteins results in increased expression of the oncogene *c-MYC* [30,58,59].

The PAM signaling pathway regulates various cellular processes, including metabolism, motility, proliferation, growth, and survival [60]. Intracellular signaling in the PAM pathway has been described in detail by Glaviano et al. [61] and Yang et al. [62]. Hyperactivation of the PAM pathway is observed in many cancers and contributes to accelerated cancer initiation and progression and the development of therapeutic resistance [63,64]. PAM promotes epithelial-mesenchymal transition (EMT) and metastasis [65,66]. Mutations in *PIK3CA*, *PTEN*, and *AKT1* have been identified in the PAM pathway of ATC. *PIK3CA,* which encodes the p110α catalytic subunit, is frequently mutated in various cancers and is known to be associated with cell signaling, proliferation, invasion, and cancer development [67,68,69]. *PIK3CA* mutations were detected in less than 20% of ATC [29,31]; however, they were more frequent than they were in DTC or PDTC [28,30]. *PTEN* is a representative tumor suppressor gene similar to *TP53* and is frequently mutated in various cancers. PTEN plays a critical role in regulating cell growth, proliferation, survival, migration, and invasion [70]. *PTEN* mutations have been demonstrated to activate the PAM pathway with a loss of function [71,72]. Additionally, loss of PTEN function in combination with alterations in *TP53* has been reported to accelerate tumor progression [73]. *PTEN* mutations in ATCs were less than 20%. Their frequency was higher than that of PTCs and PDTCs and similar to that of FTCs [28,30]. AKT1 regulates key processes such as glucose metabolism, apoptosis, cell proliferation, transcription, and cell migration. However, the *AKT1* gene mutation is rarer than that of *PI3KCA* and *PTEN* in ATC and accounts for less than 10% of mutations in most reports [28,31]. Although genetic mutations in *AKT* are rare, overactivation has been observed in many cancers, including ATC, resulting in tumorigenesis, growth, invasion, and drug resistance [74].

Other mutations were observed in Wnt-β-catenin pathway-related genes (*CTNNB1, AXIN1,* and *APC*) [75] and epigenetic-related genes such as SWI/SNF chromatin remodeling complex (*ARID1A, SMARCB1*, and *PBRM1*) and histone methyltransferases (*KMT2A, KMT2C, KMT2D,* and *SETD2*). Moreover, mutations in the *cyclin-dependent kinase inhibitor 2A (CDKN2A)* gene that encodes p16 have been observed in a few ATC cases [28,29]. Furthermore, anaplastic lymphoma kinase (ALK) mutations and fusions that activate both MAPK and PAM pathways have been detected at low rates [28,29]. Similarly, receptor tyrosine kinase (RET) fusions have been observed [76].

Although the details are discussed in Section 3.4, “Targeted Therapy,” many drugs specific to these targets are currently under development and in clinical trials. Among these, combination therapy with a BRAF inhibitor and MEK inhibition has exhibited favorable antitumor effects in patients with ATC and the BRAF V600E mutation [10].

## 3. Treatment of ATC

ATC is difficult to treat as it is aggressive, spreads rapidly within the neck, and possesses the potential to metastasize to distant body sites. After considering the available therapies, comorbidities, and patient wishes, physicians decide whether to provide aggressive treatment or palliative care based on staging and prognosis. In general, radical treatment is difficult to achieve, and multidisciplinary treatment combining surgery, radiotherapy, and chemotherapy is used to prolong life and maintain QOL, except for a few cases (Figure 2). In retrospective studies, patients who received this multimodal therapy exhibited better prognoses than those who did not [17,23,77,78,79]. Radioactive iodine (RAI) therapy and thyroid-stimulating hormone (TSH) suppression that are commonly used to treat DTC are not effective for ATC [7]. In recent years, targeted therapy has been used with a few successes in cases with specific genetic mutations. Additionally, immune checkpoint inhibitors have been used successfully in a few cases.

In this section, we describe each treatment modality’s current status and future perspectives, considering the results of representative recent studies (Table 2) and ongoing clinical trials (Table 3) of multimodal treatments.

### 3.1. Surgery

A complete surgical excision followed by adjuvant therapy is the best approach to curing ATC [15]. However, curative resection is currently beneficial in early stage cases (IVA and part of IVB), as it may lead to long-term survival [86], whereas surgery must be carefully approved to maintain QOL in cases where the long-term prognosis is not expected [87]. Specifically, R0/R1 resection can be expected if the tumor does not extend beyond the common carotid artery, whereas tumors that extend beyond the common carotid artery are often inoperable. R0 resection, also known as curative resection, indicates a microscopically margin-negative resection with no residual tumor grossly or microscopically in the primary tumor. R1 resection indicates that all macroscopic lesions are removed, but the microscopic margins are tumor-positive [88,89]. Debulking surgery that minimizes postoperative QOL is considered valuable [90,91]; however, the extent to which it affects prognosis remains unclear. The National Comprehensive Cancer Network recommends total thyroidectomy with therapeutic lymph node dissection (R0/R1) and local radiotherapy if resectable, and tracheostomy with steroids only if more strongly indicated [9,92]; However, as recommended by the American Thyroid Association, it may be practical to perform surgery for local control while taking care not to interfere with other available palliative approaches, including radiation and systemic therapy, and not to compromise QOL [7].

The role of surgery in the treatment of locally advanced and metastatic ATC is currently the subject of considerable debate, with successful cases reported in patients with ATC harboring BRAF V600E mutations who received preoperative dabrafenib plus trametinib and ultimately underwent surgery. Looking to the future, this suggests that the use of surgery may be re-evaluated, even in advanced stages.

### 3.2. Radiotherapy

Rapid local progression and recurrence of ATC are associated with the aggressive nature of the disease, and local control using surgery or external beam radiotherapy (EBRT) is important. EBRT is a treatment in which X-rays are delivered to the tumor from high-energy radiotherapy equipment located outside of the body. Radiotherapy is recommended, as EBRT is used as a preoperative or postoperative adjuvant therapy and has been demonstrated to improve the median OS in retrospective studies [18,54,93,94]. For example, in a retrospective analysis of 496 patients with ATC, Saeed et al. reported that the survival of 375 patients who received adjuvant EBRT was significantly longer than that of 121 patients who did not receive adjuvant EBRT (12.3 vs. 9.1 months) [95]. The total dose was an important factor in EBRT. Most studies have indicated that OS and local control can be predicted by the total dose [96] that is better managed with irradiation greater than 45 Gy. For example, in a National Cancer Database analysis of 1,288 patients with unresected stage IVB and IVC ATC, Pezzi et al. reported that 1-year OS rates were improved in patients treated with 60–75 Gy compared to those treated with less than 60 Gy (31% vs. 16%) [97]. In a retrospective analysis by Fan et al., delivering 60 Gy or more was significantly associated with a lower risk of local progression and longer OS (10.6 vs. 3.6 months) [80]. Although the conventional irradiation regimen of 2 Gy once daily, as recommended by the American Thyroid Association and National Comprehensive Cancer Network guidelines, has been used as the standard option, hyperfractionated, accelerated, and fractionated irradiation have been studied. Hyperfractionated and accelerated irradiation have been considered to overcome the rapid progression and radioresistance of ATC. While certain reports suggest improved local control [98,99,100], others conclude that it neither reduces toxicity nor improves outcomes [101,102], ultimately leading to controversy. In contrast, hypofractionated irradiation, which delivers a higher dose in a shorter time, is used to improve quality of life and local control. Several clinical trials have exhibited promising results using hypofractionated RT for the treatment of ATC [103,104]. However, as discussed by Oliinyk et al. [102], limited data are evaluating hypofractionated regimens, and their use in the actual treatment of ATC remains conclusive. Irradiation techniques play an important role. The thyroid gland is located near the spinal cord, and this makes it difficult to avoid the spinal cord and deliver high doses to localized areas of cancer. One technology that makes this possible is intensity-modulated radiation therapy (IMRT). IMRT is a technique that delivers a high dose to the tumor while minimizing the dose to organs at risk (OAR) by varying the intensity of the radiation in the field during treatment, thus allowing precise dose delivery even to tumors with complex shapes. IMRT can precisely deliver radiation to thyroid cancer cells while reducing the radiation dose to the spinal cord [105]. IMRT is recommended as the standard of care for radiation therapy in ATC due to its advantages in reducing local recurrence, toxicity, and treatment complications [7,8]. Recently, volumetric modulated arc therapy has been developed, in which the device is rotated at different speeds and dose rates to modulate intensity, ultimately resulting in shorter treatment times and improved treatment accuracy [106]. Altogether, it is currently accepted that the appropriate treatment is to utilize IMRT to achieve a total dose of 60 Gy or more at 2 Gy per dose. Radiotherapy is often used in combination with chemotherapy, molecular targeted therapy, and immunotherapy to treat ATC. The clinical trials of combination therapies, including radiotherapy, are shown in Table 4.

However, photon beam radiotherapy exhibits limitations regarding delivering a radical dose to the tumor site while accounting for damage to the OAR. From a future perspective, one way to overcome this problem is to use particle beam radiotherapy such as proton beam radiotherapy (PBRT) and carbon ion radiotherapy (CIRT). Particle beams enter the body with low energy when they enter the surface, transfer most of their energy to a certain depth that is called the Bragg peak, and then decay rapidly. These physical characteristics allow particle beams to deliver higher doses to cancers while protecting the OAR compared to radiation therapy using photon beams. The relative biological effectiveness (RBE) of a proton beam is 1.1- to 1.2-fold higher than that of X-rays [109], whereas that of carbon ion beams is 2- to 3-fold higher than that of X-rays [110]. As higher RBE values indicate greater cell-killing capacity, particle beams are more effective for cancer treatment than X-rays [111]. The only actual application of particle therapy for ATC was reported by Youssef et al. [112]. Patients with ATC with recurrence after thyroidectomy and radioiodine therapy were treated with two cycles of intensity-modulated PBRT (IMPT) with QUAD shot radiation (3.7 Gy delivered in four fractions twice daily at least 6 h apart for 2 consecutive days, repeated every 4 weeks, and with concurrent chemotherapy with doxorubicin and dacarbazine). At the last follow-up at 12 months, the disease had not progressed locally or systemically, thus indicating that PBRT was effective for treating ATC. Carbon ion beams are even more biologically effective than proton beams and thus exert more promising therapeutic effects. Recently, intensity-modulated CIRT (IMCT) was developed using pencil beam scanning technology, in which the total dose is made uniform by the sum of the individual beams, but the intensity of each beam is different as in IMRT [113]. IMCT can further reduce the dose of irradiation to neighboring OARs compared to that of conventional CIRT. Although not included in ATC, there have been several reports detailing the results of IMCT in head and neck cancer that indicate a favorable therapeutic effect and a low incidence of acute and late toxicity [114,115]. Multi-ion radiotherapy (MIRT) is currently under development. Moreover, it combines ion beams with different linear energy transfers (LETs) of noncarbon ions with carbon ion beams, such as higher LET beams (e.g., oxygen and neon ions) targeting areas of resistance and lower LET beams (e.g., helium ions) for the boundary with normal tissue near the tumor. This approach offers a promising new treatment option for patients with complex cancers [116]. Additionally, the higher LET beams used in MIRT are more effective against radioresistant hypoxic cells than are carbon ions, thus potentially leading to earlier tumor reoxygenation [117,118,119].

### 3.3. Chemotherapy

Anthracyclines such as doxorubicin, platinum-based drugs such as cisplatin and carboplatin, and taxanes such as paclitaxel and docetaxel are the main chemotherapeutic drugs used for ATC [81]. However, the effects of these agents on ATCs are moderate and transient and are often disappointing. These guidelines recommend the rapid use of paclitaxel + carboplatin, docetaxel + doxorubicin, paclitaxel alone, or docetaxel alone when there are no therapeutic targets without molecular abnormalities or when targeted therapies are not available [7,8]. They can be used as bridging chemotherapeutic agents before the use of molecularly targeted agents [82]. Among these, taxanes have been reported to be effective for the treatment of ATC. For example, Higashiyama et al. treated stage IVB and IVC patients with ATC and weekly PTX induction chemotherapy and observed complete (CR) and partial response (PR) in 8% and 23% of patients, respectively [120]. They reported that the OS rate of paclitaxel-treated patients with stage IVB disease was better than that of patients who did not receive chemotherapy or those who received drugs other than paclitaxel. However, it is challenging to treat ATC with anticancer drugs alone.

Therefore, a future strategy is to use them in combination with other modalities. For example, taxanes increase the percentage of radiosensitive G2/M phase cells; as a result, their combination with radiotherapy may be effective. A report evaluated the use of docetaxel in combination with radiotherapy in patients with ATC and observed that 67% of patients achieved CR and the remaining 33% achieved PR, although the number of patients was small [121]. They reported that 83% of patients were alive at a median follow-up of 21.5 months. These findings suggest that a combination of taxane-based chemotherapy and radiotherapy may be effective for patients with ATC.

### 3.4. Targeted Therapy

Recently, genome sequencing analysis revealed molecules that are frequently mutated in ATCs and are involved in tumor progression and malignant transformation. Therapeutic agents targeting these molecules have been investigated. These agents target receptor tyrosine kinases (RTK) and downstream molecules of the MAPK and PAM signaling pathways.

#### 3.4.1. Inhibitors of MAPK Pathway

Inhibitors of the MAPK pathway are potential agents for cancer therapy, and many compounds have been identified in clinical and preclinical trials. Several RAF and MEK inhibitors have been approved; however, no ERK inhibitors have yet been approved.

Dabrafenib is a BRAFV600E kinase inhibitor, and trametinib is an MEK inhibitor. Their combination therapy has been approved by the U.S. Food and Drug Administration (FDA) for the treatment of patients with ATC and BRAF V600E mutations. This is due to resistance to BRAF inhibitors that develops within months of BRAF blockade alone [122]. Several mechanisms have been proposed to induce this resistance, including the expression of splicing variants of the BRAF gene [123], inhibition of the negative feedback mechanism in the MAPK pathway [124], and activation of integrin/FAK signaling [125]. In vitro studies revealed that dabrafenib and trametinib inhibit ATC cell proliferation [126]. Subbiah et al. reported that in a phase II study evaluating the efficacy of dabrafenib plus trametinib in 36 patients with ATC, the CR and PR were 8% and 47%, respectively, and the median OS was 14.5 months [10]. In a recently reported UK clinical trial, CR and PR were achieved in 12% and 71% of patients with ATC, respectively, with a median OS of 6.9 months [127].

Another combination of BRAF and MEK inhibitors includes encorafenib (BRAF inhibitor) and binimetinib (MEK inhibitor). In a phase II study of 22 patients with thyroid cancer, including five patients with ATC and BRAF mutation-positive ATC with local invasion or distant metastasis, that evaluated the efficacy of the combination of encorafenib and binimetinib, the overall response rate (ORR) was 80% (CR, one case; PR, three cases) [128].

Vemurafenib is a selective BRAF inhibitor approved by the FDA for the treatment of patients with metastatic melanoma harboring the BRAF V600E mutation [129]. Zhang et al. reported that vemurafenib inhibits tumor growth in an in vivo xenograft mouse model using an ATC cell line [130]. A clinical trial evaluating the efficacy of vemurafenib in patients with cancer due to the BRAF V600E mutation, including seven patients with ATC, indicated that the percentages of patients with cancer achieving CR and PR were 14% and 14%, respectively [131]. Clinical trials are currently evaluating the efficacy of combining vemurafenib with cobimetinib, a MEK inhibitor, in BRAF-positive thyroid cancers, including ATC.

Other drugs, such as PLX8394 (BRAF inhibitor) and selumetinib (MEK inhibitor), have been investigated for the treatment of thyroid cancer [132]. Additionally, the ERK inhibitor DEL-22379 has been reported to decrease cell viability and inhibit the metastasis of ATC cells with BRAF mutations in vitro and in vivo [133].

#### 3.4.2. Inhibitors of PAM Pathway

Molecules involved in this signaling pathway have received considerable attention in recent years, and many drugs targeting them have been studied and evaluated in animals and humans.

Everolimus inhibits the mTOR complex 1 (mTORC1). Owonikoko et al. reported that everolimus is effective against ATC cells both in vitro and in vivo [134]. However, the results of clinical trials investigating the efficacy of everolimus have been disappointing [83,135,136]. mTOR exists in two functionally and structurally distinct complexes, mTOR complex 1 (mTORC1) and mTORC2. Inhibition of mTORC1 inhibits ribosome biogenesis and suppresses cell proliferation, whereas inhibition of mTORC2 activates AKT and promotes cell growth [137]. This may have been due to the limited efficacy of everolimus.

In contrast, ATP-competitive dual mTORC1/2 inhibitors such as MLN0128 [138], AZD2014 [139,140], and AZD8055 [141] have been developed that can efficiently block the PAM pathway without causing negative feedback induction of mTORC2. Among these compounds, a phase II clinical trial is currently underway to evaluate the efficacy of MLN0128 for the treatment of metastatic ATC.

Other compounds, including CUDC-907 (PI3K and HDAC inhibitor) [142], copanlisib (PI3K inhibitor) [143], and buparlisib (PI3K inhibitor) [144], have been studied in the context of thyroid cancer.

#### 3.4.3. RTK Inhibitors

RTKs are transmembrane glycoproteins on the cell surface that regulate cell proliferation, differentiation, and survival by activating several important intracellular signaling pathways, such as the MAPK and PAM pathways. RTKs include vascular endothelial growth factor receptor (VEGFR), epidermal growth factor receptor (EGFR), fibroblast growth factor receptor (FGFR), cluster of differentiation 117 (c-KIT), rearranged during transfection (RET), and platelet-derived growth factor receptor (PDGFR). Overexpression and activation of mutations in cancer cells have been reported, and drugs targeting them have been developed.

Lenvatinib is an inhibitor of VEGFR, FGFR, c-kit, RET, and PDGFR, and it is approved by the FDA and the European Medicines Agency for the treatment of iodine-131 (I-131) refractory DTC [145]. Ferrari et al. reported that lenvatinib suppresses the proliferative capacity of ATC cells in vitro and in vivo [146]. In a clinical trial of patients with ATC, Takahashi et al. reported a median OS of 10.4 months and an ORR of 24% in the lenvatinib group [84]. Reports have suggested that lenvatinib provides clinical benefits to patients with advanced ATC [147,148]. Although a high response rate can be expected, there is a risk of a shortened prognosis in patients with tracheal or carotid artery invasion due to fistula formation. Furthermore, lenvatinib alone is not an effective treatment for ATC [149] and has been disappointing in terms of prolonging the survival of unresectable ATC [150].

Similar to lenvatinib, single-agent multi-kinase inhibitors (MKIs), including sorafenib, gefitinib, imatinib, sunitinib, and pazopanib, have exhibited efficacy in vitro and in vivo [151,152,153,154,155,156,157] but have not exhibited promising results in clinical trials [107,158,159,160,161,162,163]. These MKIs are not effective when prescribed alone for the treatment of ATC but may be potentially useful in combination with other targeted therapies. Furthermore, the therapeutic effects of TRK or RET inhibitors are expected to be effective when mutations or fusions of TRK or RET are observed.

Targeted drugs have been developed against various target molecules, and a few of them, such as dabrafenib plus trametinib and encorafenib plus binimetinib, have demonstrated therapeutic efficacy. However, the emergence of drug resistance within a few months after a successful response has become a problem. Overcoming drug-resistant cancer is a major challenge in the development of molecularly targeted cancer therapies. Currently, there are no concrete, effective means to solve this problem.

Future prospects include elucidating the mechanisms of drug resistance at the basic research level and, in clinical practice, identifying effective therapeutic agents by studying the mutations that occur when drug resistance develops through genetic testing.

### 3.5. Immunotherapy

Tumor cells grow and metastasize through various mechanisms to escape recognition and attack by the immune system. Tumor-induced immunosuppression occurs in two ways. The first occurs through the induction of immunosuppressive cells that accumulate around the tumor and secrete immunosuppressive factors. Another mechanism involves the expression of immunosuppressive molecules and their receptors, such as programmed death ligand/programmed death-1 (PD-L1/PD-1), galectin-9/TIM-3, LAG-3, and CTLA-4. These are known as immune checkpoints that inhibit the activation of effector T lymphocytes, ultimately leading to the immune escape of the tumor. A recent trend in immunotherapy is the inhibition of these immune checkpoints and the restoration of immune function.

PD-L1 expressed on tumor cells binds to PD-1 on T cells and suppresses the proliferation and function of T cells, thereby weakening their immune response to tumor cells. PD-L1 expression in cancer cells may be primarily expressed or induced in the cancer microenvironment by the stimulation of inflammatory cytokines produced by tumor-infiltrating lymphocytes (TILs) and tumor-associated macrophages (TAMs) surrounding tumor cells [164]. PD-L1 is highly expressed in ATC [85,165,166], and Cantara et al. reported that immunohistochemistry using two PD-L1 antibodies exhibited high positivity rates of 65% and 90%, respectively [167]. Therefore, anti-PD-L1 and anti-PD-1 antibodies may be effective therapies for ATC. In contrast, CTLA-4 expressed on T cells binds to B7 molecules (CD80 and CD86) on antigen-presenting cells and inhibits T cell activation. CTLA-4 is expressed in regulatory T cells (Tregs). When Tregs bind to antigen-presenting cells, T cells cannot bind to antigen-presenting cells and cannot be activated. Therefore, anti-CTLA-4 antibodies are expected to be used for cancer therapy.

There are reports of the use of the anti-PD-1 antibodies spartalizumab and pembrolizumab and the anti-PD-L1 antibody atezolizumab in the treatment of ATC. In a phase II study by Capdevila et al. involving 42 patients with ATC treated with spartalizumab intravenously at a dose of 400 mg every 4 weeks, the ORR was 19% (CR, 7%; PR, 12%), and it was 29% in patients with PD-L1 expression and even higher in patients with high PD-L1 expression. The median OS was 5.9 months, with 40% of patients surviving for 1 year [85]. In a retrospective study, Iyer et al. demonstrated the efficacy of pembrolizumab in combination with lenvatinib, trametinib, or a combination of dabrafenib and trametinib in patients with ATC [168]. Moreover, there were no CRs, but 42% of patients achieved OR. The median OS after the addition of pembrolizumab was 6.93 months. Wang et al. investigated six patients with ATC and BRAF V600E mutation who underwent complete surgical resection and received dabrafenib plus trametinib with or without pembrolizumab [169]. No local recurrence was detected in any patient. Four patients with pembrolizumab achieved CR within the observation period (7.8–26.0 months), while two patients without pembrolizumab died due to distant metastasis. These results suggest that the combination of dabrafenib and trametinib with ICI would be more effective for patients with ATC and the BRAF V600E mutation. A phase 2 pilot study is ongoing to evaluate the efficacy of dabrafenib and trametinib in combination with cemiplimab, an anti-PD-1 antibody, in the context of ATC treatment. Moreover, reports suggest that a combination of lenvatinib and pembrolizumab may be effective for treating ATC [170]. A study evaluating the efficacy of lenvatinib in combination with nivolumab is ongoing in Japan (jRCT2080224758), and results are expected in the near future. While immunotherapy is successful in a few cases, it can cause side effects. Immune checkpoint inhibitors reactivate immune cells that have been suppressed by cancer cells. These side effects are known as immune-related adverse events (irAEs) and have been reported to include interstitial pneumonia, colitis, type 1 diabetes, endocrine disorders such as thyroid dysfunction, liver and kidney dysfunction, skin disorders, myasthenia gravis, myositis, and uveitis [171,172]. Chronic side effects have been reported, such as those that persist for more than three months after the end of treatment with immune checkpoint inhibitors [173].

The irAEs are becoming very complicated due to the combination of immunotherapy and other therapies. In the future, it is necessary to move toward establishing and managing a safety management system across medical departments and professions in collaboration with multiple professions to address these issues.

### 3.6. Others

Recently, it has been increasingly recognized that microbiota, including those in the gut and oral cavity, influence the pathogenesis, prognosis, and treatment response of various diseases, including cancer [174]. Several studies have reported a relationship between thyroid cancer and the microbiota. Research on the relationship between oral bacteria has reported that an increase in the genera *Alloprevotella*, *Anaeroglobus*, and *Acinetobacter*, unclassified *Bacteroidales*, and unclassified *Cyanobacteriales* was observed in the saliva of patients with thyroid cancer [175,176]. Moreover, studies exist on the relationship between gut microbiota and thyroid cancer. The relative abundances of *Neisseria* and *Streptococcus* were significantly higher in the gut flora of patients with thyroid cancer than those in healthy controls [177]. A study examining patients with PTC revealed that the abundance of *Firmicutes* was higher in the fecal flora than it was in healthy participants [178].

Previously, the thyroid gland was believed to be a sterile organ; however, recent technological advances have revealed that it is colonized by microorganisms. Microorganisms exist in thyroid cancer tissues, and a relationship between bacterial flora and disease status has been reported. One study has demonstrated a correlation between *Sphingomonas* abundance within the tumor microenvironment and lymph node metastasis [179]. A study examining patients with PTC revealed a high abundance of *Proteobacteria* in the tumors. Additionally, tumor bacterial diversity increased in patients with a higher T stage [178]. Regarding the therapeutic response to postoperative I-131 treatment for PTC, low levels of the genera Dorea and Bifidobacterium were associated with a poor prognosis [180]. However, there have been no reports investigating the relationship between ATC and the bacterial flora. Treatment methods for ATC, such as radiotherapy, chemotherapy, and immunotherapy, are associated with bacterial flora [181]. Therefore, in the future, microflora will attract attention as a target for new treatment strategies for ATC, and further research and development are expected [182].

### 3.7. Best Supportive Care

Although promising therapies have been developed for the treatment of ATC in recent years, it remains a difficult-to-cure cancer with a high mortality rate. The patient wishes to be fully considered when deciding whether to pursue aggressive treatment, including palliative care for pain and dyspnea and hospice care [183].

## 4. Conclusions

Although the treatment of ATC is evolving and diversifying, the optimal treatment has not yet been identified, ultimately resulting in a poor prognosis. The rapid progression of ATC necessitates immediate diagnosis and treatment. A multidisciplinary care team, including endocrinologists, oncologic surgeons, medical oncologists, radiation oncologists, radiologists, and palliative care staff, must be formed to determine the treatment plan and provide appropriate care as soon as possible while considering patient wishes. Recent studies have demonstrated that BRAF-mutant ATCs exhibit dramatically improved prognosis. However, the problem of drug resistance and treatment strategies for wild-type BRAF remain underexplored. A better understanding of the molecular biology of ATCs is expected to lead to the development of novel targeted therapies. Although radiotherapy remains the mainstay of ATC treatment, recent technological developments such as intensity-modulated irradiation and particle therapy are remarkable. The successful use of combinations of these novel drugs and therapies may lead to the development of treatment strategies to improve the ATC prognosis. Further clinical trials using many patients are required.

## Figures and Tables

**Figure 1 biomedicines-12-01286-f001:**
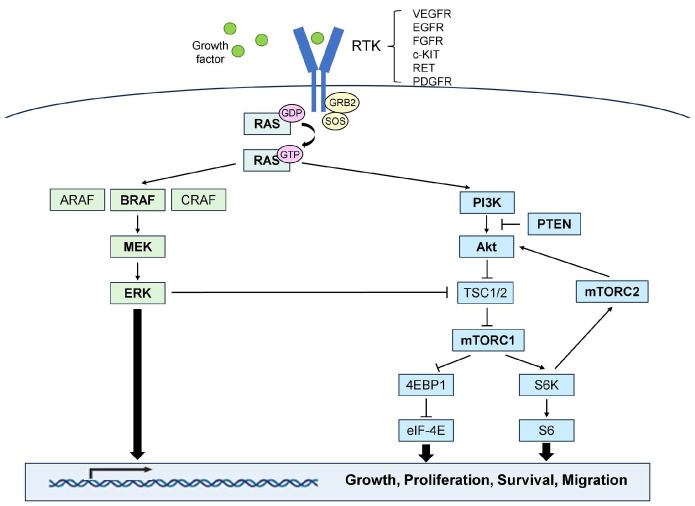
Schematic illustration of the MAPK signaling and PAM signaling pathways in the ATC. Sharp arrows (→) and blunt arrows (┤) indicate promotion and inhibition, respectively.

**Figure 2 biomedicines-12-01286-f002:**
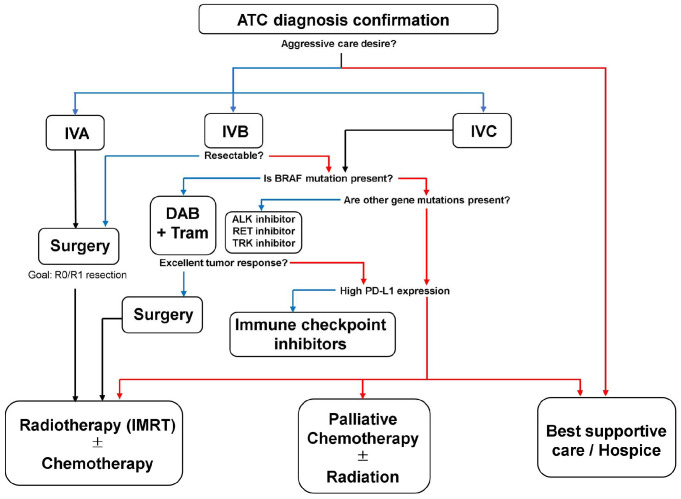
Initial treatment of stages IVA, IVB, and IVC ATC. Taken from [7,12], with a few modifications. For each question, blue arrows indicate “Yes,“ and red arrows indicate “No.“ R0: resection for cure or complete remission. R1: complete gross resection; tumor removed grossly but not microscopically.

**Table 1 biomedicines-12-01286-t001:** Staging and TMN classification of ATC according to the AJCC eighth edition.

Stage	T Category		N Category		M Category	
IVA	T1:	Tumor ≤ 2 cm in greatest dimension limited to the thyroid	N0:	No metastasis to regional nodes	M0:	No distant metastasis
	T2:	Tumor > 2 cm but ≤4 cm in greatest dimension limited to the thyroid				
	T3a:	Tumor > 4 cm limited to the thyroid				
IVB	T1		N1:	Metastasis to regional nodes	M0	
	T2					
	T3a					
	T3b:	Gross extrathyroidal extension invading only strap muscles (sternohyoid, sternothyroid, thyrohyoid, or omohyoid muscles) from a tumor of any size	N0–N1		M0	
	T4a:	Gross extrathyroidal extension invading subcutaneous soft tissues, larynx, trachea, esophagus, or recurrent laryngeal nerve from a tumor of any size				
	T4b:	Gross extrathyroidal extension invading prevertebral fascia or encasing a carotid artery or mediastinal vessels from a tumor of any size				
IVC	T1–T4		N0–N1		M1:	Distant metastasis

**Table 2 biomedicines-12-01286-t002:** Recent studies with multimodal treatment in ATC.

Treatment	Study	Number of Patients	Number of Patients					Median OS (Months)	ORR (%)	Reference
			Surgery	RT	CTx	MTT	IMM			
RT (66 Gy) + Doxorubicin (10–20 mg/m^2^ weekly) or Paclitaxel (80 mg/m^2^ weekly)	Retrospective	104 IVA:5 IVB:76 IVC:23	52	101	99	0	0	7	N.D.	Fan et al. (2020) [80]
Doxorubicin + Docetaxel (20 mg/m^2^ weekly, each)Carboplatin + Paclitaxel (50 mg/m^2^ weekly, each)Doxorubicin only (20 mg/m^2^ weekly) Cisplatin only (30 mg/m^2^ weekly)	Retrospective	30 IVA:2 IVB:22 IVC:6 ND:5	27	30	30	0	0	21	63	Prasongsook et al. (2017) [81]
Dabrafenib (150 mg twice daily) + Trametinib (2 mg once daily)	Phase 2	36 IV:1 IVC:35	30	30	15	36	4	14.5	56	Subbiah et al. (2022) [10]
Dabrafenib (150 mg twice daily) + Trametinib (2 mg once daily)	Retrospective	16 IVB:4 IVC:12	8	7	9	16	0	9.3	50	Iyer et al. (2018) [82]
Everolimus (10 mg daily)	Phase 2	7 IVC:7	5	4	3	7	0	4.6	14	Hanna et al. (2018) [83]
Lenvatinib (24 mg daily)	Phase 2	17 IV:17	14	9	7	17	0	10.6	24	Takahashi et al.(2019) [84]
Spartalizumab (400 mg every 4 weeks)	Phase 1/2	42 IV:42	28	30	25	4	42	5.9	19	Capdevila et al. (2020) [85]

RT: radiotherapy, CTx: chemotherapy, MTT: molecular targeted therapy, IMM: immunotherapy, OS: overall survival, ORR: overall response rate.

**Table 3 biomedicines-12-01286-t003:** Ongoing clinical trials in patients with ATC.

ClinicalTrials.gov Identifier	Intervention/Treatment	Phase	Status
NCT04552769	Abemaciclib	Phase 2	Active, not recruiting
NCT05453799	Vudalimab	Phase 2	Recruiting
NCT04171622	Lenvatinib + Pembrolizumab	Phase 2	Recruiting
NCT03975231	Dabrafenib + Trametinib + IMRT	Phase 1	Recruiting
NCT05119296	Pembrolizumab (Keytruda)	Phase 2	Recruiting
NCT04420754	AIC100 CAR T Cells	Phase 1	Recruiting
NCT03449108	Aldesleukin (IL2) + Autologous Tumor Infiltrating Lymphocytes LN-145 or LN-145-S1	Phase 2	Active, not recruiting
NCT03246958	Nivolumab + Ipilimumab	Phase 2	Active, not recruiting
NCT04675710	Pembrolizumab + Dabrafenib + Trametinib + Surgery + IMRT	Phase 2	Recruiting
NCT03181100	Atezolizumab + Chemotherapy (Cobimetinib, Nab-paclitaxel, Paclitaxel, Vemurafenib)	Phase 2	Active, not recruiting
NCT03085056	Trametinib + Paclitaxel	Early Phase 1	Active, not recruiting
NCT04238624	Dabrafenib + Trametinib	Phase 2	Recruiting
NCT04759911	Selpercatinib + Surgery	Phase 2	Recruiting
NCT06007924	Avutometinib + Defactinib	Phase 2	Recruiting
NCT02041260	Cabozantinib	Phase 2	Unknown
NCT04579757	Surufatinib + Tislelizumab	Phase 1/2	Active, not recruiting
NCT05059470	Pembrolizumab + IMRT	Phase 2	Recruiting

**Table 4 biomedicines-12-01286-t004:** Clinical trials related to radiotherapy in patients with ATC.

ClinicalTrials.gov Identifier	Intervention/Treatment	Phase	Status	Reference
NCT03565536	Sorafenib + Surgery + EBRT	Phase 2	Completed	
NCT05659186	Tislelizumab + Anlotinib + RT	Phase 2	Recruiting	
NCT01236547	IMRT + Paclitaxel + Pazopanib	Phase 2	Completed	Sherman et al. (2023) [107]
NCT03122496	Durvalumab + Tremelimumab +SBRT	Phase 1	Completed	Lee et al. (2022) [108]
NCT03211117	Docetaxel + Doxorubicin + IMRT + Pembrolizumab + Surgery	Phase 2	Completed	
NCT04675710	Pembrolizumab + Dabrafenib + Trametinib + Surgery + IMRT	Phase 2	Recruiting	
NCT03975231	Dabrafenib + Trametinib + IMRT	Phase 1	Recruiting	
NCT00004089	Chemotherapy (Fluorouracil, Hydroxyurea, Paclitaxel) + Surgery + RT	Phase 2	Completed	
NCT00077103	Chemotherapy (Cisplatin, Doxorubicin) + Fosbretabulin + RT	Phase 1/2	Terminated	
NCT05059470	Pembrolizumab + IMRT	Phase 2	Recruiting	

RT: radiotherapy, SBRT: stereotactic body radiotherapy.

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
