# Peer review of "Recent Trends and Potential of Radiotherapy in the Treatment of Anaplastic Thyroid Cancer"

_biomedicines, 2024, doi:10.3390/biomedicines12061286_

Round 1
Reviewer 1 Report
Comments and Suggestions for Authors
Well written manuscript.
It is imperative that all abbreviations are expanded just before the first use of the abbreviation. For example, DXR might mean a different drug in different countries.
It may be worthwhile to summarise the results of each modality of treatment as a table, giving the author, year of publicaiton, number of patients included, stage at inclusion, regimen chosen, randomisation if any, the parameter being measured for response, as well as the p value if any, outlining the difference between the randomised arms.
Author Response
We appreciate the highly constructive reviewers’ comments on our manuscript and believe that we have been able to address them fully, as detailed below.
Reply (Answer; A) to Reviewer’s comments and questions (Comment; C):
Reviewer Comments:
Reviewer 1
C1: Well written manuscript. It is imperative that all abbreviations are expanded just before the first use of the abbreviation. For example, DXR might mean a different drug in different countries. It may be worthwhile to summarise the results of each modality of treatment as a table, giving the author, year of publication, number of patients included, stage at inclusion, regimen chosen, randomisation if any, the parameter being measured for response, as well as the p value if any, outlining the difference between the randomised arms.
A1: We have corrected the abbreviations as you suggested. Also, since you pointed out that the abbreviations for drugs are different in each country, we decided not to use any abbreviations to avoid making a mistake.
Reviewer 2 Report
Comments and Suggestions for Authors
The present article provides a comprehensive summary of the existing treatment methods for anaplastic thyroid carcinoma, a highly aggressive and challenging disease. Despite the difficulties posed by its radical nature, the article emphasizes the potential benefits of employing multidisciplinary approaches, such as surgery, radiotherapy, and chemotherapy, to extend life expectancy and maintain the quality of life for patients. The authors acknowledge that conventional therapies, including radioactive iodine and thyroid-stimulating hormone suppression, have proven effective up to a certain extent. Furthermore, the authors contend that the study's findings will pave the way for the development of innovative therapies that integrate targeted molecular agents and immunotherapy.
Section 1+2
• More examples to directly substantiate the fact that genetic mutations lead to functional repercussions in ATC pathogenesis are included in this section.
Considering this, even though the segment pays much emphasis on the multidisciplinary approach, it would have been better if some proven case studies had been included and proven to have succeeded through the approach.
Section 3
• Although this section provides a general outlook of the treatment options available, it could have been expanded to present a more critical analysis and discussion regarding the limitations of current therapies. For example, the problematic issues of drug resistance to targeted therapy and potential side effects of immunotherapy could have been addressed.
• The incorporation of more recent research studies and clinical trials will provide this section with more scientific validity and hence allow readers to access the latest, most relevant information.
• This section could have been further elaborated with ongoing research and future directions in the treatment of ATC, which would provide the reader with a general idea of the ATC field at present.
Specific comments:
• The manuscript is comprehensive in providing an overview of the treatment options for ATCs, but there is still a need to detail the technical terms and abbreviations used. For instance, terms like "R0/R1 resection" and "EBRT" should be defined in an attempt to make the document more accessible for someone not previously exposed to the field.
2. Strengths of Ideas Presented
• This manuscript focuses on challenges in the treatment of ATC. However, to strengthen this argument, the authors should have been able to provide a more critical analysis of the effectiveness and limitations of each treatment modality. It would have been a more balanced viewpoint to include data on the survival rates and quality of life outcomes associated with each of these options.
3. Argument Coherence
• It is logically structured, with every subsection dealing with some aspect of treatment, yet clear transitions between the subsections would improve the flow of ideas. Subheadings or introductory sentences signaling new topics would make it easier to read.
4. Areas that
• The introduction and discussion can be substantially enriched by the inclusion of more recent studies and current clinical trials for scientific validity. It would be more updated if studies in the last few years were referred to.
• A more critical analysis of current therapies, including challenges such as drug resistance in targeted therapy or the potential side effects of immunotherapy, would provide a more nuanced perspective.
• Discussion of ongoing research efforts and future directions for ATC treatment could expand the scope of this manuscript. Future clinical trials and new therapies will lead to potential advances in the field. • Provide explanations or definitions of technical terms and abbreviations to enhance accessibility so that readers not acquainted with the field can understand the content.
Author Response
We appreciate the highly constructive reviewers’ comments on our manuscript and believe that we have been able to address them fully, as detailed below.
Reply (Answer; A) to Reviewer’s comments and questions (Comment; C):
C1:
Section 1+2
- More examples to directly substantiate the fact that genetic mutations lead to functional repercussions in ATC pathogenesis are included in this section.
Considering this, even though the segment pays much emphasis on the multidisciplinary approach, it would have been better if some proven case studies had been included and proven to have succeeded through the approach.
A1: We wrote it in the Targeted Therapy or Immunotherapy section, but as you pointed out, we mentioned the case studies in Sections 1 and 2 (Page 1, Lines 41-44 and Page 5, Lines 192-195).
C2:
Section 3
- Although this section provides a general outlook of the treatment options available, it could have been expanded to present a more critical analysis and discussion regarding the limitations of current therapies. For example, the problematic issues of drug resistance to targeted therapy and potential side effects of immunotherapy could have been addressed.
- The incorporation of more recent research studies and clinical trials will provide this section with more scientific validity and hence allow readers to access the latest, most relevant information.
- This section could have been further elaborated with ongoing research and future directions in the treatment of ATC, which would provide the reader with a general idea of the ATC field at present.
A2: We have modified it better to describe current therapies' limitations and future prospects. In particular, we have addressed your issues, such as drug resistance in targeted therapies and potential side effects in immunotherapy. In addition, we have added Table 2 for the latest clinical trials being conducted for ATCs.
C3: Specific comments:
- The manuscript is comprehensive in providing an overview of the treatment options for ATCs, but there is still a need to detail the technical terms and abbreviations used. For instance, terms like "R0/R1 resection" and "EBRT" should be defined in an attempt to make the document more accessible for someone not previously exposed to the field.
C3: We have followed your suggestion and written definitions for the technical terms. (Page 7, lines 224-227, 244-245, and page 8, lines 274-275)
C4: Strengths of Ideas Presented
- This manuscript focuses on challenges in the treatment of ATC. However, to strengthen this argument, the authors should have been able to provide a more critical analysis of the effectiveness and limitations of each treatment modality. It would have been a more balanced viewpoint to include data on the survival rates and quality of life outcomes associated with each of these options.
A4: We have better described the effectiveness and limitations of each treatment in regards to your comments (Page 8, lines 281-288, Page 9, lines 332-340, Page 11, lines 428-4368, and Page 12, lines 482-493).
C5: Argument Coherence
- It is logically structured, with every subsection dealing with some aspect of treatment, yet clear transitions between the subsections would improve the flow of ideas. Subheadings or introductory sentences signaling new topics would make it easier to read.
A5: We have included an introductory sentence in each section to clearly identify current problems and new perspectives on overcoming them.
C6: Areas that
- The introduction and discussion can be substantially enriched by the inclusion of more recent studies and current clinical trials for scientific validity. It would be more updated if studies in the last few years were referred to.
- A more critical analysis of current therapies, including challenges such as drug resistance in targeted therapy or the potential side effects of immunotherapy, would provide a more nuanced perspective.
- Discussion of ongoing research efforts and future directions for ATC treatment could expand the scope of this manuscript. Future clinical trials and new therapies will lead to potential advances in the field. • Provide explanations or definitions of technical terms and abbreviations to enhance accessibility so that readers not acquainted with the field can understand the content.
A6: We have added a table of recent clinical trials of molecularly targeted therapies for Anaplastic thyroid cancer as Table 2. Based on your suggestions, we have noted problems with current therapies and discussed future directions that could improve them. We have also added explanations and definitions of technical terms and abbreviations.
Reviewer 3 Report
Comments and Suggestions for Authors
The review of the manuscript titled "Current Developments in the Treatment of Anaplastic Thyroid 2 Cancer" has raised some concerns regarding the quality of the content. The lack of important meta-analysis information and an unfocused introduction section have been noted. Additionally, the document contains typos, minor mistakes, and language-related problems that reduce the overall credibility of the review. As a result, the research community may not find it of great interest or significance.
Furthermore, the authors did not include a table of trials on molecular target therapies used in Anaplastic Thyroid Cancer treatment, which is a significant omission. It is worth noting that a similar review has already been published in other journals with DOIs 10.1177/15330338231169870, 10.1097/MED.0000000000000823, and 10.1186/s13044-020-00091-w. As a result, researchers who expect Biomedicines (impact factor 4.7) to publish articles and reviews about state-of-the-art research may not find this review valuable.
In conclusion, despite the authors' effort, the review presented in the manuscript is too superficial and does not provide any valuable or significant information about the drug research field. Therefore, it may be better suited for another journal that places less emphasis on innovation.
Comments on the Quality of English LanguageModerate editing of English language required
Author Response
We appreciate the highly constructive reviewers’ comments on our manuscript and believe that we have been able to address them fully, as detailed below.
Reply (Answer; A) to Reviewer’s comments and questions (Comment; C):
C1: The review of the manuscript titled "Current Developments in the Treatment of Anaplastic Thyroid 2 Cancer" has raised some concerns regarding the quality of the content. The lack of important meta-analysis information and an unfocused introduction section have been noted. Additionally, the document contains typos, minor mistakes, and language-related problems that reduce the overall credibility of the review. As a result, the research community may not find it of great interest or significance.
Furthermore, the authors did not include a table of trials on molecular target therapies used in Anaplastic Thyroid Cancer treatment, which is a significant omission. It is worth noting that a similar review has already been published in other journals with DOIs 10.1177/15330338231169870, 10.1097/MED.0000000000000823, and 10.1186/s13044-020-00091-w. As a result, researchers who expect Biomedicines (impact factor 4.7) to publish articles and reviews about state-of-the-art research may not find this review valuable.
In conclusion, despite the authors' effort, the review presented in the manuscript is too superficial and does not provide any valuable or significant information about the drug research field. Therefore, it may be better suited for another journal that places less emphasis on innovation.
A1: At your suggestion, we have added a table of recent clinical trials of molecularly targeted therapies for treating undifferentiated thyroid cancer as Table 2. The introduction has been rewritten, and the entire article has been proofread in English. As noted in the comments, there are many other review articles on targeted therapy, but we have focused on the potential of radiotherapy. We believe that the review article discussing the application of particle therapy to ATC is, to the best of our knowledge, unique in its perspective.
Round 2
Reviewer 1 Report
Comments and Suggestions for Authors
Thank you for the correction based on the observations
The article would be more readable if the information is tabulated as suggested
Author Response
We appreciate the highly constructive reviewers’ comments on our manuscript and believe that we have been able to address them fully, as detailed below.
Reply (Answer; A) to Reviewer’s comments and questions (Comment; C):
Reviewer Comments:
Reviewer 1
C: Thank you for the correction based on the observations
The article would be more readable if the information is tabulated as suggested.
A: Thank you very much for your comments. We have summarized the results of recent studies that represent the text in a table (Table 2).
Reviewer 3 Report
Comments and Suggestions for Authors
I noticed that there have been some significant changes made to the revised manuscript in response to the reviewers' comments. If the authors believe that the review article should focus on targeted therapy, but we have instead emphasized the potential of radiotherapy, they will need to change the article title to reflect this shift in focus. Additionally, they should include a table containing clinical trial information specifically related to radiotherapy.
Comments on the Quality of English LanguageModerate editing of English language required.
Author Response
We appreciate the highly constructive reviewers’ comments on our manuscript and believe that we have been able to address them fully, as detailed below.
Reply (Answer; A) to Reviewer’s comments and questions (Comment; C):
Reviewer Comments:
C: I noticed that there have been some significant changes made to the revised manuscript in response to the reviewers' comments. If the authors believe that the review article should focus on targeted therapy, but we have instead emphasized the potential of radiotherapy, they will need to change the article title to reflect this shift in focus. Additionally, they should include a table containing clinical trial information specifically related to radiotherapy.
A: We appreciate your suggestion; we have changed the title to "Recent Trends and Potential of Radiotherapy in the Treatment of Anaplastic Thyroid Cancer" to emphasize the potential of radiotherapy in treating ATC. We have also included a table of clinical trial information related to radiotherapy in the article as Table 4.
Round 3
Reviewer 3 Report
Comments and Suggestions for Authors
The authors made significant revisions, and based on the reviewer's comments, the MS can be acceptable for publication.
Comments on the Quality of English LanguageMinor editing of English language required